# Change in Physical Activity after Diagnosis of Diabetes or Hypertension: Results from an Observational Population-Based Cohort Study

**DOI:** 10.3390/ijerph16214247

**Published:** 2019-11-01

**Authors:** Matthias Rabel, Filip Mess, Florian M. Karl, Sara Pedron, Lars Schwettmann, Annette Peters, Margit Heier, Michael Laxy

**Affiliations:** 1Department of Sport and Health Sciences, Technical University of Munich, Munich 80992, Germany; filip.mess@tum.de; 2Helmholtz Zentrum München–German Research Center for Environmental Health (GmbH), Institute of Health Economics and Health Care Management, Garching 85748, Germany; florian.karl@helmholtz-muenchen.de (F.M.K.); sara.pedron@helmholtz-muenchen.de (S.P.); lars.schwettmann@helmholtz-muenchen.de (L.S.); michael.laxy@helmholtz-muenchen.de (M.L.); 3German Center for Diabetes Research (DZD), Munich-Neuherberg 85764, Germany; 4Helmholtz Zentrum München—German Research Center for Environmental Health (GmbH), Institute of Epidemiology, Neuherberg 85764, Germany; peters@helmholtz-muenchen.de (A.P.); heier@helmholtz-muenchen.de (M.H.); 5KORA Study Centre, University Hospital Augsburg, Augsburg 86156, Germany; 6Rollins School of Public Health—Global Diabetes Research Center, Emory University, Atlanta, GA 30322, USA

**Keywords:** physical activity, diagnosis, diabetes, hypertension, health behavior, chronic disease

## Abstract

*Background:* Chronic diseases like diabetes mellitus or hypertension are a major public health challenge. Irregular physical activity (PA) is one of the most important modifiable risk factors for chronic conditions and their complications. However, engaging in regular PA is a challenge for many individuals. The literature suggests that a diagnosis of a disease might serve as a promising point in time to change health behavior. This study investigates whether a diagnosis of diabetes or hypertension is associated with changes in PA. *Methods:* Analyses are based on 4261 participants of the population-based KORA S4 study (1999–2001) and its subsequent 7-and 14-year follow-ups. Information on PA and incident diagnoses of diabetes or hypertension was assessed via standardized interviews. Change in PA was regressed upon diagnosis with diabetes or hypertension, using logistic regression models. Models were stratified into active and inactive individuals at baseline to avoid ceiling and floor effects or regression to the mean. *Results:* Active participants at baseline showed higher odds (OR = 2.16 [1.20;3.89]) for becoming inactive after a diabetes diagnosis than those without a diabetes diagnosis. No other significant association was observed. *Discussion:* As PA is important for the management of diabetes or hypertension, ways to increase or maintain PA levels in newly-diagnosed patients are important. Communication strategies might be crucial, and practitioners and health insurance companies could play a key role in raising awareness.

## 1. Introduction

Chronic diseases like cancer, diabetes mellitus, or cardiovascular conditions are considered to be among the most important challenges to public health. Further, chronic diseases are the main driver for impaired quality of life, and account globally for more than 70% of deaths [1]. Nonetheless, they are preventable. The key risk factors for chronic diseases include poor diet, alcohol consumption, smoking, and physical inactivity [2]. Furthermore, even though the benefits of positive health behavior changes are well known [3], people often have difficulties initiating a change in their health behavior for several reasons [4].

The literature suggests that so-called “health shocks” might serve as a “natural nudge” [5], “wake-up call” [6], “window of opportunity” [7] or “teachable moment” [8] to initiate a health behavior change. A health shock in this context is considered to be an unpredicted illness that lowers a person’s health status [9]. Other sources define health shocks as an outbreak of a disease [5] or as a negative effect on someone’s health [10]. Further studies expand the term and describe a health shock as a change in self-assessed health [10] or a decline in grip strength [11]. Finally, according to Agüero and Beleche [5], the diagnosis of an illness like diabetes or hypertension can also be interpreted as a health shock [5]. Within this study, we follow Agüero and Beleche’s [5] definition. Therefore, a medical diagnosis might be a promising point in time to initiate health behavior changes, as physicians occupy a central role when it comes to health behavior [12].

There are several theoretical health behavior theories supporting the hypothesis that a diagnosis might lead to a health behavior change. One very popular theoretical model to describe behavior changes is the Health Action Process Approach Model by Schwarzer. This psychological approach suggests that risk perception is an important factor when it comes to building an intention to change a health behavior [13]. The Protection Motivation Theory by Rogers is a further theory proclaiming threat appraisals as a fundamental component of behavior change [14]. Both the Health Action Process Approach and the Protection Motivation Theory offer a theoretical foundation for the event of a diagnosis to be viewed as an important factor explaining health behavior and health behavior changes.

Empirically, a few studies have investigated possible associations between a diagnosis of several kinds of diseases and a change in different kinds of health behaviors. Whereas the effects of different types of diagnoses, for example a decline in self-assessed health status, cancer diagnosis, cardiovascular conditions, stroke, and diabetes, on smoking cessation have been documented by multiple studies [6,7,10,15,16,17,18], the body of evidence for other behaviors is smaller, and study results are more heterogeneous. For example, it was reported that alcohol consumption decreased slightly after the diagnosis of one of multiple chronic diseases [6] or after a cancer diagnosis [19]. Regarding dietary behavior, it was shown that people eat a healthier diet after a cancer [15] or a diabetes diagnosis [20]. 

In terms of physical activity (PA), studies reported decreasing levels of PA after a cancer diagnosis [15,19], or no change after experiencing a diabetes diagnosis [20,21] or any multiple chronic disease diagnosis [6]. In contrast, one other study reported significant increases in women’s PA after a diabetes diagnosis [22].

Relative to the few studies on associations between changes in PA after a diabetes diagnosis, the number of studies examining PA changes after a hypertension diagnosis is even smaller; only two studies have dealt with the latter relationship. Hernandez et al. described that only 25% of inactive participants increased their PA after a hypertension diagnosis [23]. Neutel and Campell reported small but non-lasting improvements in PA behavior after a hypertension diagnosis [24].

For both, i.e., diabetes and hypertension, the benefits of PA are well known [25,26], and regular PA is recommended in treatment guidelines [27,28]. Hence, knowledge about a possible health behavior change after a diabetes or hypertension diagnosis is crucial for a couple of reasons concerning public health. First, it yields important information on designing public health interventions. It might strengthen the claim that for the whole process of behavior change, the moment of receiving information about health consequences could be even more important than the knowledge itself of the harmful consequences [5]. Second, as evidence suggests that physicians play an important role when it comes to promoting behavioral changes [12], behavioral interventions promoted by physicians could be designed more effectively in the context of the diagnosis. Third, a positive health behavior change could lower the risk of disease reoccurrence, increase health-related quality of life, and positively influence longevity [6]. 

Consequentially, the aim of this paper is to investigate possible associations between a diabetes or hypertension diagnosis and PA change. For this, we use well pheno-typed data from a large German population-based cohort study comprising three measurements over a period of 14 years.

## 2. Materials and Methods 

### 2.1. Study Design and Participants

The data originated from the KORA (Cooperative Health Research in the Region of Augsburg) S4/F4/FF4 cohort study located in Southern Germany. The S4 (1999–2001) study was a population-based cross-sectional health survey including 4261 adult participants (67% response rate of all eligible participants) aged 25 to 74 years [29]. Of those, 3080 (80% response rate) took part in the first follow-up F4 (2006–2008) examination and 2279 (79% response rate) in the second follow-up FF4 (2013–2014) examination (compare caption in Figure 1 for details). For all studies, information on participants was collected during standardized medical examinations, standardized face-to-face interviews conducted by trained medical staff, and self-administered questionnaires. Additional information on the S4 study regarding the sampling methods and data collection has been published elsewhere [30]. All study participants gave written informed consent. The Ethics Committee of the Bavarian Medical Association approved the KORA S4/F4/FF4 studies.

### 2.2. Measures

Physical activity: Changes in PA was the main outcome of this study. Participants’ levels of PA were assessed during standardized interviews using two separate, four-category item questions reflecting weekly time spent on leisure-time PA during winter and summer. Within the KORA research platform, both items were combined into a single variable with the following categories: (1) “(almost) no activity”, (2) “about 1 h per week, irregularly”, (3) “about 1 hour per week, regularly” and (4) “regularly, more than 2 h per week”. The questions on leisure-time PA were derived from the German Cardiovascular Prevention Study conducted between 1979 and 1995. The questions were validated in the KORA population [31] using a PA diary as comparison.

For the present study, PA was dichotomized by condensing categories (1) and (2), representing irregular PA and inactive individuals, and categories (3) and (4), representing regular PA and active individuals. We decided to draw the line between categories (2) and (3), as (3) represents regular PA, and can therefore be considered as habitualized behavior, whereas category (2) still symbolizes irregular and infrequent PA behavior. This cut-off for the dichotomization of PA was also used in previous KORA studies [32]. The change in PA is a binary variable indicating whether a person showed a change in PA at a follow-up session.

Incident diabetes and hypertension diagnosis: To consider individual awareness of a diabetes or hypertension diagnosis, we used self-reported information on the existence of diabetes (Questionnaire item: “Are you diabetic?”, and further information on the intake of antidiabetics) and hypertension (Questionnaire item: “Have you ever had elevated blood pressure or been diagnosed with high blood pressure?”) from the standardized interviews. An incident diagnosis of diabetes or hypertension was coded binary if the condition was not present at baseline but present at follow up.

Covariates: Other relevant variables assessed at all three measurement points were sex (male, female), levels of secondary education (lower, intermediate, higher; corresponding to German “Hauptschule”, “Realschule”, “Gymnasium”), family status (living alone, living together), age (continuous), body mass index (BMI) (continuous), and physical and mental health-related quality of life (both continuous). Both components of health-related quality of life were measured using the SF-12 health survey [33].

### 2.3. Statistical Analyses

We investigated the association between diabetes or a hypertension diagnoses and changes in PA between two measurement points. Therefore, participants with existing diabetes or hypertension were excluded from the respective models. To increase the statistical power, the two seven-year follow-up periods from S4 to F4 and from F4 to FF4 were combined, comprising the observations of both periods. This means that we treated the S4-F4 follow-up period and the F4-FF4 follow-up period as two separated periods. Consequently, participants with an incident diagnosis in the first follow-up period only appeared once in the final analysis sample, whereas participants without an incident diagnosis in the first follow-up period appeared twice. To account for dependencies due to the repeated measurement of participants, the participant ID was set as a random intercept in the models. To avoid ceiling or floor effects and regression to the mean, we stratified our sample according to baseline PA categories, resulting in an “inactive” and “active” stratum. Thus, for individuals being inactive at baseline, we compared the odds of becoming active between those with an incident diagnosis of diabetes or hypertension and those without such a diagnosis. Conversely, for active individuals at baseline, we compared the odds of becoming inactive between those with a diagnosis of diabetes or hypertension and those without such a diagnosis. Figure 1 provides an overview of the study population and the analysis set-up. Four logistic regression models were fit in order to investigate possible changes in PA after experiencing a diabetes or hypertension diagnosis. For both strata, i.e., active and inactive, changes in PA were regressed on either a diabetes diagnosis or a hypertension diagnosis. The models were adjusted for sex, age, education, family status, baseline BMI, change in BMI, and change in physical and mental health-related quality of life. For a better convergence, all continuous covariates were standardized, with mean equals zero and standard deviation equals one. 

All data-related processes and statistical analyses were conducted using RStudio (Version 1.1.456, RStudio Inc., Boston, USA) [34]. Logistic regression models were carried out using the “lme4” package (Version 1.1.20) [35].

## 3. Results

### 3.1. Descriptive Analyses

Considering the S4 baseline sample with 4261 participants, 29 participants (0.68%) had missing values for PA, resulting in a baseline sample of 4232 participants of whom 51% were female and for whom the mean age was 49.2 ± 13.9 years. The mean BMI was 27.2 ± 4.7 kg/m^2^. According to the baseline PA value, 2058 participants showed active leisure-time PA and 2174 showed inactive leisure-time PA. Table 1 depicts descriptive statistics for the baseline S4 study.

Regarding diabetes, there were 166 participants with existing diabetes at S4, 214 with existing diabetes at F4, and 219 with existing diabetes at FF4. Regarding hypertension, there were 1079 participants with existing hypertension at S4, 1043 with existing hypertension at F4, and 831 with existing hypertension at FF4. The present study focuses on a new incident diagnoses of diabetes and hypertension. Therefore, Figure 2 shows the number of incident cases of diabetes and hypertension by PA changes between S4 and F4 (columns 1 & 2), between F4 and FF4 (columns 3 & 4), and for the dataset of the assembled S4/F4 and F4/FF4 periods (columns 5 & 6). The assembled data set is the one used for the main logistic regression models. 

### 3.2. Associations between Diabetes or Hypertension Diagnosis and PA Change

Figure 3 shows the odds ratios for changing PA after a diabetes or hypertension diagnosis. For participants who were active at baseline, the figure shows the odds of becoming inactive, while for inactive participants at baseline, it shows the odds of becoming active after receiving a diabetes or hypertension diagnosis. The results from the logistic regression models show that participants who were active at baseline had higher odds (OR = 2.16, 95% CI = [1.20,3.89]) of changing PA and becoming inactive after facing a diabetes diagnosis compared to individuals who did not receive such a diagnosis in the same period. Regarding hypertension, the odds of the already-active participants to become inactive after the diagnosis were smaller than those for participants that did not receive a diagnosis (OR = 0.93, 95% CI = [0.64,1.35]). For those participants who were inactive at baseline, the odds of becoming active after a diabetes diagnosis were smaller than those for participants who were not diagnosed with diabetes (OR = 0.74, 95% CI = [0.46,1.18]). The odds of participants who were inactive at baseline to become active after a hypertension diagnosis were the same as those for participants that did not receive a hypertension diagnosis (OR = 1.02, 95% CI = [0.75,1.40]). A complete overview of the four logistic regression models, including all variables, can be seen in Appendix A in the supplement. 

### 3.3. Additional Analyses

We ran several sensitivity analyses to check the robustness of our results. In an additional analysis, we changed the decision line for the dichotomization of PA. In that case, we only classified the category “(almost) no activity” as inactive, and combined the other three categories as active. The association between the two assessed diagnoses and change in PA did not differ from the main analyses. The results of the models, including the altered PA variable, are presented in Appendix A in the supplement.

In another additional analysis, we ran the same logistic regression models on each follow-up period separately. These models showed similar results, justifying our approach to combine both follow-up periods in the main analyses. The corresponding results are presented in detail in Appendix A in the supplement.

## 4. Discussion

We examined the associations between a diabetes or hypertension diagnosis and changes in PA to investigate whether a diagnosis can be seen as catalyst for health behavior changes. For this, we analyzed the data of 4232 participants, including three measurement points over the course of 14 years. The results showed no indication for improved PA among participants with a diabetes or hypertension diagnosis compared to participants without a diagnosis. Initially active participants showed higher odds of changing their PA and becoming inactive after a diabetes diagnosis. All other results had small effect sizes and were not statistically significant. Active participants did not change their PA after a hypertension diagnosis. Inactive participants prior to a diabetes or hypertension diagnosis were not more likely to become active than participants without a diabetes or hypertension diagnosis. We performed several sensitivity analyses showing the robustness of our findings.

### 4.1. Interpretation and Implications

Although the literature suggests a pathway from the onset of a chronic disease leading to a change in health behavior, we have to point out that it might as well be the other way around. It might also be possible that a lack of PA leads to the onset of a chronic disease. With our study, we can only investigate associations between an diabetes or hypertension diagnoses and PA changes, but we cannot imply a causal relationship between those parameters. Hence, although we follow a theoretical framework that suggests the plausibility of the effect of a diabetes or hypertension diagnosis on PA, and although our longitudinal models were adjusted for important confounders, our estimates do not imply causality, and should rather be interpreted as descriptive measures. 

According to the treatment guidelines for diabetes and hypertension, general practitioners should stress the importance and benefits of regular PA [27,28]. Yet, in our study, we did not see an improvement in PA. Moreover, regarding diabetes diagnoses, we observed a higher decline in PA compared to people without a diabetes diagnosis. 

One possible reason for this could be that people might not be aware of the benefits of modified PA behavior in the context of a diabetes or hypertension diagnosis. Other studies report that overweight or obese participants do not attribute their weight to a greater health risk [36,37]. This lack of knowledge concerning the connection of excessive weight and health could also be applicable to PA.

Furthermore, the aforementioned awareness problem might also explain deviations from the Health Action Process Approach model and the Protection Motivation Theory. The perceived future risk of a diabetes or hypertension diagnosis may have too little effect on the theory’s risk perception and threat appraisal. The high prevalence of diabetes and hypertension, as well as good treatment options, might be possible reasons for a limited perceived risk. Several studies have reported the limited use of risk perception as a predictor of behavioral changes within, for example, the framework of the Health Action Process Approach Model [38,39,40].

Another interpretation could be that the patient might be more accustomed to medical treatment than to changes in PA as a proper reaction to a medical diagnosis. Jarbøl et al. suggest that people might prefer a change in their lifestyle behaviors over medicinal treatment when the perceived effects of both options on disease management are equal [41]. Considering that the advantages of regular PA are not immediately noticeable [20,42], people might prefer a medicinal therapy after a diabetes or hypertension diagnosis.

Emotional factors might serve as another explanation for non-improved or even declined levels of PA. Thomas et al. reported that a lack of confidence in the ability to exercise and the fear of deteriorating diabetes are major barriers to physical activity for people with diabetes [43].

In view of these interpretations, there is potential to better communicate the benefits of regular PA to the public. There is a need for better information about the positive effects of behavioral changes in the context of secondary prevention. Public health initiatives should aim to strengthen health literacy, and research should focus on ways in which to effectively communicate the benefits of positive health behavior to the patients and the public. Following the clinical practice guidelines for diabetes and hypertension, increased PA needs to be considered as an important factor in health care management which ought to be better reimbursed by insurers. The importance of adherence to recommended PA levels, especially regarding secondary prevention, has to be stressed by practitioners, and counselling time should be adequately reimbursed by health insurance companies.

### 4.2. Comparison with Similar Studies

Several studies have mentioned positive changes in health behavior following a diagnosis. When comparing these studies, one has to acknowledge the different characteristics of different diseases and their effect on PA. Chronic diseases like cancer, stroke, or myocardial infarction come along with serious physical impairments that often make it difficult to stay physically active, whereas diabetes or hypertension usually do not restrict people in terms of their physical activity levels. It seems that besides the covered types of diagnoses, the results vary for different health behaviors. While several studies have reported a positive effect on smoking behavior [7,10,15,16,17,18], the present study did not show comparably positive effects on PA behavior. In their qualitative study, Dolor et al. found that practitioners consider counseling on PA and weight management to be more time-consuming than counseling on smoking [44].

Compared to previous studies, our results are in line with the majority of the published results reporting no increase in PA after a diabetes diagnosis [6,20]. Leung et al. reported that the observed higher rates of exercise initiation for participants that have been diagnosed with diabetes did not reach statistical significance [21]. In their study, including exclusively women aged from 50 to 79 years, Schneider, et al. observed an increase in PA after a diabetes diagnosis compared to women without a diagnosis [22]. Divergence with the results of Schneider et al. might be due to the different study population and a different operationalization of change in PA. Studies on the association between a hypertension diagnosis and PA change are scarce. Neutel and Campbell reported no lasting changes in lifestyle behavior after a hypertension diagnosis apart from smoking cessation [24]. Hernandez et al. mentioned that only a quarter of inactive participants started to increase their PA after a diabetes diagnosis [23].

### 4.3. Limitations

The present study has several strengths, including a large population-based sample and a long study period with three measurement points. Further, the study addresses an important public health issue for which only scarce, heterogeneous evidence is available. However, the following limitations should be considered when interpreting the study’s results. We have no data on the time of the diagnosis. Thus, we cannot disentangle the timing of a diagnosis and PA change, and do not know if a diagnosis preceded PA change or vice versa. A study investigating the relationship between diabetes and changes in PA found a decrease in PA over time after the diagnosis. Similar to our study, Chong et al. found no significant difference regarding PA changes for people that had been diagnosed with diabetes and those who had not [20]. A further limitation of the present study is the assessment method for PA. Information on PA behavior was self-reported and was gathered in standardized interviews. Therefore, measurement errors and recall bias could be an issue. However, one has to balance the tradeoff between accuracy and feasibility, especially in large cohort studies with many variables, such as the KORA study. Further, the considerably reduced cohort size over the course of the study period could result in biased effects, as healthier and more active people are more likely to remain in the cohort. Admittedly, this could only be considered a bias if, for example, people with a diabetes diagnosis and PA level increase were more likely stay in the cohort than those with a diagnosis but with a decrease in PA levels. Finally, although we adjusted our models for numerous potential confounders, our results still might be affected by additional confounders. Possible further confounders could be age-related comorbidities or prescribed medication for prediabetes or prehypertension. Therefore, residual confounding cannot be ruled out. 

## 5. Conclusions

In conclusion, the present study showed that individuals who practice regular PA are more likely to become inactive after a diabetes diagnosis compared to individuals who did not receive a diabetes diagnosis. Instead, for the same group, a hypertension diagnosis was not associated with changes in PA. Furthermore, a diabetes or hypertension diagnosis is not associated with improved PA levels among inactive individuals. This indicates that the suggested “open window” following a clinical diagnosis in which patients are more likely to adopt health behavioral changes may not exist, or may not yet have been effectively exploited for diabetes and hypertension. In order to make better use of this potential in the future, one should emphasize the general awareness of the benefits of PA in the context of secondary prevention. 

## Figures and Tables

**Figure 1 ijerph-16-04247-f001:**
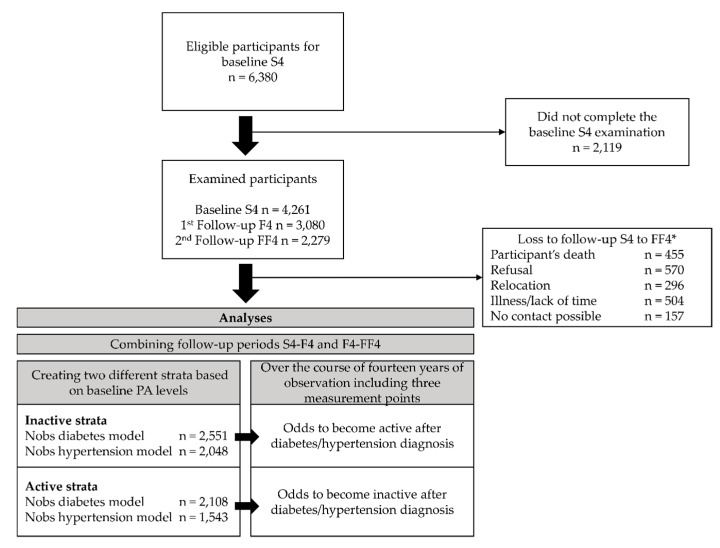
Study population and analysis flow chart. * Of the baseline S4 sample, 174 died, 204 moved outside the study region or to an unknown location, and 12 refused to be further contacted. Of the remaining 3871 eligible participants, 176 could not be contacted, 220 were too ill or busy, and 395 refused to participate further. Thus, 3080 participants (80% response rate) took part in the first follow-up F4 (2006-2008) examination. Of the F4 sample, 168 died, 97 moved outside the study region or to an unknown location, and 67 refused to be contacted further. Of the resulting eligible 2748 participants, 48 could not be contacted, 332 were too ill or busy, and 207 refused to participate further. Adding 118 participants from S4 without F4 information, 2279 (79% response rate, excluding 118 participants without F4 information) participants took part in the second follow-up FF4 (2013–2014) examination. Annotations: *n* = number of participants, Nobs = number of observations.

**Figure 2 ijerph-16-04247-f002:**
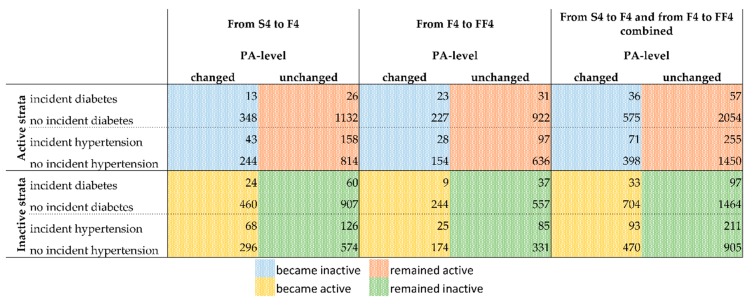
Overview of the cases with incident diagnosis of diabetes or hypertension for the active and inactive strata by PA change. See Appendix A in the supplement for a detailed overview of how the presented numbers were derived. Numbers of observations for the combination of both follow-up periods: Active-Diabetes = 2722; Active-Hypertension = 2174; Inactive-Diabetes = 2298; Inactive-Hypertension = 1679. Deviations from the numbers of Figure 2 and Figure 3 are due to missing values in covariates for which the models of Figure 3 were adjusted. Abbreviations: PA = physical activity; S4 = baseline study (2000); F4 = first follow-up (2007); FF4 = second follow-up (2014).

**Figure 3 ijerph-16-04247-f003:**
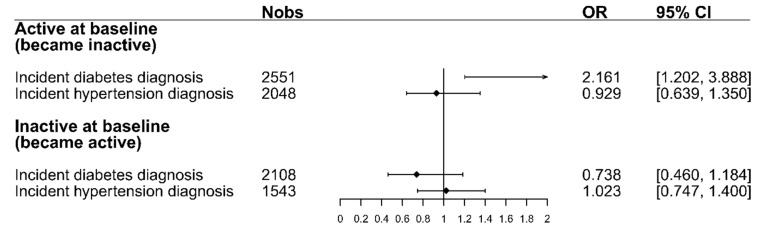
The odds-ratios for changing PA after a diabetes/hypertension diagnosis for both baseline PA strata. ORs for the active stratum display the odds of changing to inactive. ORs for the inactive stratum display the odds of changing to active. All models were adjusted for the following factors: sex, age, education, family status, baseline BMI, change in BMI, and change in physical and mental health-related quality of life. Abbreviations: Nobs = Number of observations, OR = Odds ratio, 95% CI = 95% Confidence interval.

**Table 1 ijerph-16-04247-t001:** Descriptive information on the baseline S4 study for the total sample.

Parameter	Baseline S4	
N	4232	
Female	2158	0.51
Age (mean) (SD)	49.18	13.94
BMI (mean) (SD)	27.22	4.73
Physical activity		
Inactive	1447	0.34
About 1 h/week irregularly	727	0.17
About 1 h/week regularly	1200	0.28
More than 2 h/week	858	0.20
Levels of secondary education		
Lower	2287	0.54
Intermediate	981	0.23
Higher	963	0.23
Family status		
Single, living alone	483	0.11
Single, living together with partner	223	0.05
Married, living together	2901	0.69
Married, separated	79	0.02
Divorced	296	0.07
Widowed	250	0.06
Present diagnosis (Excluded from respective models)		
Diabetes	166	0.04
Hypertension	1079	0.26

Absolute and relative frequencies regarding relevant covariates. Abbreviations: N = number of observations, SD = standard deviation, h/week = hours per week.

## Data Availability

The data that support the findings of this study are available from KORA (https://www.helmholtz-muenchen.de/en/kora/for-scientists/cooperation-with-kora/index.html) but restrictions apply to the availability of these data, which were used under license for the current study, and so are not publicly available. However, data can be requested through an individual project agreement with KORA via the online portal KORA.passt (https://epi.helmholtz-muenchen.de/).

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
