# Peer review of "Change in Physical Activity after Diagnosis of Diabetes or Hypertension: Results from an Observational Population-Based Cohort Study"

_ijerph, 2019, doi:10.3390/ijerph16214247_

Round 1
Reviewer 1 Report
The article is well structured, has a sound scientific question and appropriate methods. The conclusion is supported by the results.
I only have one suggestion: would it be possible that a decrease in PA between the baseline and follow ups is actually related to the onset of diabetes?
The way that the study question was formulated doesn't allow for this interpretation, but this makes more sense to me than believing that people reduced PA after diabetes diagnosis. It is only a suggestion for a point that could be raised in the discussion.
Reviewer 2 Report
The authors provide new evidence for the impact of significant clinical diagnoses with respect to following lifestyle changes - in this case: physical activity. The introduction provides sufficient information about the available literature and current base of knowledge.
In an observational cohort study (KORA), the authors select follow-up intervals in order to assess changes in physical activity semi-quantitatively. Also, they evaluate incident cases of type 2 diabetes and hypertension. As main outcome, they calculate odds ratios for changes in PA when comparing subjects with / without incident diagnosis of T2DM or hypertension. This methodological approach is suitable to gain new insights into the research question. Statistical adjustment is done for a variety of variables, however, as in most cohort studies, relevant co-factors may have been left unconsidered.
A decline in PA can possibly attributed to age-related illnesses, which are not covered by adjustment for age alone: arthrosis of the lower limbs, neuropathy, cancer, ... Also, some medication is prescribed more often in T2DM, even shortly before diagnosis. Some of these treatments may impair the possible walking distance or even the will to leave the home due to side effects (metformin --> diarrhea; gliflozins and diuretics --> sudden and frequent diuresis; antihypertensive drugs --> vertigo, fainting, tiredness). Also, depression is more common in T2DM and may affect spontaneous PA.
It is surprising, that a significant effect is seen for T2DM but not hypertension, although both disorders carry similarities: lack of specific symptoms, tight relation to the same lifestyle factors, variable magnitude of severity within short time periods. Thus, both disorders can be missed in clinical practice, when a sufficient screening approach is not conducted. For hypertension, regular RR measurements are well established at all general practitioners. On the other hand, T2DM is often diagnosed with long delay, often in the course of another acute illness (severe infection, myocardial infarction, stroke). This difference in outpatient screening coverage may explain and the difference in the effects and should be investigated, also.
Please clarify, if adjustment for the aformentioned factors is possible.
Legend for Fig. 3 should include the information for statistical adjustment.
The above mentioned co-factors should be discussed in the final section of the manuscript.
Reviewer 3 Report
In "Change in physical activity after diagnosis of diabetes or hypertension: Results from an observational population-based cohort study," Dr. Rabel and colleagues present their investigation based on a long-term observational study on physical activity and diabetes/hypertension diagnoses.
There are a number of questions regarding the statistical analyses, and these must be answered/resolved before this manuscript is considered further.
In the following, I will only refer to Diabetes models, but the same set of questions are applicable to Hypertension models.
1) Please define 'incident cases' and 'prevalent cases'. Line 160: "Participants with prevalent cases ... were excluded from ... models" Does this mean that those with Diabetes=Yes at baseline were not included in the diabetes models?
1a) How many had both diabetes and hypertension at baseline, therefore not a part of this research. Are they included in Table 1?
2) I could not reconstruct the numbers shown in Figure 2 from the information given in lines 184 to 188. Please clarify and explain how/if the numbers like 166, 214, 219 can be computed from the total number of participants and other numbers in Figure 2. I am a little confused about numbers in Figure 2. Is 36+57+575 +2054 the final sample size for the active-diabetes model? How is that related to the overall n=4232?
3) Because there are two follow-up time points (F4 and FF4) diabetes=Yes/No were observed twice (excluding baseline), I think. But I don't see that in the table in Appendix A1. Please explain. One of my biggest concerns is mixing of time-course of change in PA and diagnoses. Diagnosis at FF4 (Yes**) should not affect PA at F4 (irregular*).
| example | S4 | F4 | FF4 |
| PA | regular | irregular* | irregular** |
| Diabetes | No | Yes* | Yes** |
How does the model address this?
4) Also, how does one know that diabetes diagnosis affect PA and not vice versa? In other words, lack of PA may be a cause of newly diagnosed diabetes.
5) By including the random intercept, you are estimating the subject-specific probabilities. Aren't marginal models more appropriate for the purpose of the study?
6) Some of the models have quite small sample size (The effective sample size is the size of smaller group, not the total number of subjects), please address stability of the coefficient estimates and if that is a problem.
7) Please state assumptions regarding missingness (data attrition -decreasing number of participants as time goes on.).
8) Please state the model assumptions and if they are satisfied.
9) Did any of the models fail to converge without standardization of continuous variables?
Round 2
Reviewer 2 Report
All comments have been addressed sufficiently.